# THE EFFECTIVENESS OF PRE-TRAINED CODE EMBEDDINGS

## ABSTRACT

Word embeddings are widely used in machine learning based natural language processing systems. It is common to use pre-trained word embeddings which provide benefits such as reduced training time and improved overall performance. There has been a recent interest in applying natural language processing techniques to programming languages. However, none of this recent work uses pre-trained embeddings on code tokens. Using extreme summarization as the downstream task, we show that using pre-trained embeddings on code tokens provides the same benefits as it does to natural languages, achieving: over 1.9x speedup, 5% improvement in test loss, 4% improvement in F1 scores, and resistance to over-fitting. We also show that the choice of language used for the embeddings does not have to match that of the task to achieve these benefits and that even embeddings pre-trained on human languages provide these benefits to programming languages.

## 1 INTRODUCTION

One of the initial steps in a machine learning natural language processing (NLP) pipeline is converting the one-hot encoded $\mathbb{R}^V$ tokens into dense $\mathbb{R}^D$ representations, with $V$ being the size of the vocabulary, $D$ the embedding dimensions and $V << D$. This conversion is usually done with a single layer neural network, commonly called an embedding layer.

The parameters of the embedding layer can either be initialized randomly or initialized via "pre-trained" parameters obtained from a model such as word2vec (Mikolov et al., 2013b;a), GloVe (Pennington et al., 2014) or a language model (McCann et al., 2017; Howard & Ruder, 2018; Peters et al., 2018).

It is common to use pre-trained parameters (most frequently the GloVe embeddings), which act as a form of transfer learning (Mou et al., 2016) similar to that of using pre-trained parameters for the convolutional kernels in a machine learning computer vision task (Huh et al., 2016; Kornblith et al., 2018). These parameters in the embedding layer are then fine-tuned whilst training on the desired downstream task.

The use of these pre-trained embeddings over random initialization allows machine learning models to: train faster, achieve improved overall performance (Kim, 2014), increase the stability of their training, and reduce the amount of over-fitting (Mou et al., 2016).

Recently there has been an increased interest in applying NLP techniques to programming languages and software engineering applications (Vechev & Yahav, 2016; Allamanis et al., 2017a), the most common of which involves predicting the names of methods or variables using surrounding source code (Raychev et al., 2014; 2015; Allamanis et al., 2015; 2017b; Alon et al., 2018b;a).

Remarkably, none of this work takes advantage of pre-trained embeddings created on source code. From the example below in table 1, we can see how semantic knowledge (provided by the pre-trained code embeddings) of the method body would help us predict the method name, i.e. knowing how pi and radius are used to calculate an area and how height and width are used to calculate an aspect ratio.

```
float getSurfaceArea (int radius) {
        return 4 * Math.PI * radius * radius;
}
```

```
float getAspectRatio (int height, int width) {
        return height / width;
}
```

Table 1: Examples showing how the semantics of the variable names within a method can be used to reason about the name of the method body

This semantic knowledge is available to us as even though computers do not need to understand the semantic meaning of a method or variable name, they are mainly chosen to be understood by other human programmers (Hindle et al., 2012).

In this paper, we detail experiments using pre-trained code embeddings on the downstream task of predicting a method name from a method body. This task is known as extreme summarization (Allamanis et al., 2016b) as a method name can be thought of as a summary of the method body. Our experiments are focused on answering the following research questions:

1. **Do pre-trained code embeddings reduce training time?**
2. **Do pre-trained code embeddings improve performance?**
3. **Do pre-trained code embeddings increase stability of training?**
4. **Do pre-trained code embeddings reduce over-fitting?**
5. **How does the choice of corpora used for the pre-trained code embeddings affect all of the above?**

To answer RQ5, we gather a corpus of C, Java and Python code and train embeddings for each corpus separately, as well as comparing them with embeddings trained on natural language. We then test each of these on the same downstream task, extreme summarization, which is in Java. We also release the pre-trained code embeddings. [1]

As far as we are aware, this is the first study on the effectiveness of pre-trained code embeddings applied to an extreme (code) summarization task.

## 2 MODELS

### 2.1 LANGUAGE MODEL

We train our embeddings using a language model (LM). We choose a LM over the word2vec or GloVe models as LMs have shown to capture long-term dependences (Linzen et al., 2016) and hierarchical relations (Gulordava et al., 2018). We believe both of these properties are essential for predicting a method name from method body. The long-term dependencies are required due to the average length of the method body over 72 tokens[2]. The hierarchical relations are needed due to the way data flows through variables within the method body, starting from the method argument(s) (at the top of the hierarchy) to the return value(s) (at the bottom of the hierarchy).

A language model is a probability distribution over sequences of tokens. Each token, $x$, is represented by a one-hot vector $x \in \mathbb{R}^V$, with $V$ being the size of the vocabulary. The probability given to a sequence of tokens $x_1, ..., x_T$ can be calculated as:

$$p(x_1, ..., x_T) = \prod_{t=1}^{T} p(x_t | x_{t-1}, ..., x_1)$$

---

[1]Code and embeddings to be released at a later date.
[2]We use a token to refer to an atomic part of a sequence of code.

We model this probability distribution with a recurrent neural network trained to predict the next token in a sequence of given tokens. Specifically, we use the AWD-LSTM-LM model (Merity et al., 2017; 2018) due to its performance at modeling natural languages and open source implementation. [3]

## 2.2 CONVOLUTIONAL ATTENTION MODEL

The extreme summarization task uses the Copy Convolutional Attention Model from Allamanis et al. (2016a). Briefly, the model takes a series of code tokens from the method body, $c$, as input and outputs the code tokens that form the method name, $m$. It generates the method name one token at a time, using a recurrent hidden state, $h_t$, provided by a Gated Recurrent Unit (GRU) (Cho et al., 2014) and a series of convolutional filters over the embeddings of the tokens $c$, which produce attention (Bahdanau et al., 2014) features, $L_{feat}$. It also has a mechanism to directly copy tokens from the body to the output.

This model was chosen as it is the state-of-the-art on the extreme summarization dataset used and provided a clear improvement over baseline models. It also has an open source implementation. [4]

## 3 EXPERIMENTAL SETUP

### 3.1 EMBEDDING DATASET

The dataset used for the pre-trained code embeddings was gathered from GitHub. To ensure the quality of the data we only used projects with over 10,000 stars and manually checked each project's suitability, i.e. did not use projects which were tutorials or guides.

After scraping the appropriate projects for each of the three languages (C, Java and Python) we tokenized each, converting each token to lowercase as well as splitting each token into subtokens on `camelCase` and `snake_case`, e.g. `getSurfaceArea` becomes `get`, `surface` and `area`. This was done to match the tokenization of the extreme summarization dataset. There are approximately 100 million tokens across 20 million lines of code for each language. Each of the embeddings has their own distinct vocabulary, e.g. not all tokens that appear in the C corpus appear in the Java corpus.

For the natural language embeddings we used the WikiText-103 dataset Merity et al. (2016), as it contains a comparable 103 million tokens.

### 3.2 LANGUAGE MODEL

The AWD-LSTM-LM model was trained with all default parameters from the open source implementation, with the exception of: the embedding dimension changed to 128 and the hidden dimension changed to 512. The embedding dimension was changed to match that of the original Copy Convolutional Attention Model, and the hidden dimension was changed to fit in GPU memory.

Tokens that were not in the most 150,000 common or did not appear at least 3 times were converted into an `<unk>` token and the model was trained until the validation loss did not decrease for 5 epochs.

### 3.3 EXTREME SUMMARIZATION DATASET

The extreme summarization task dataset is detailed in Allamanis et al. (2016a). Briefly, it consists of 10 Java projects selected for their quality and diversity in application. For each project, all full Java methods are extracted with the method body used as the input and the method name used as the target. Each project has their own vocabulary, e.g. tokens that appear in one project may not appear in any others.

All tokens are formatted the same as the embedding dataset to ensure maximum vocabulary overlap between each pre-trained embeddings and each Java project.

---

[3]https://github.com/salesforce/awd-lstm-lm
[4]http://groups.inf.ed.ac.uk/cup/codeattention/

## 3.4 CONVOLUTIONAL ATTENTION MODEL

The Copy Convolutional Attention Model was trained with all default parameters from the open source implementation, and was trained for 25 epochs.

The model was trained on each project separately and was run 5 times on each project for each of the embeddings. The results were then averaged together for each project.

## 4 RESULTS

Table 2 shows the rank 1 F1 scores achieved for each of the embeddings. On average, we achieve a 4% relative improvement in F1 scores for each of the embeddings.

| Project Name | Description | Random F1 | C F1 | Java F1 | Python F1 | English F1 |
|---|---|---|---|---|---|---|
| cassandra | Distributed Database | 48.1 | 51.7 | 50.4 | 50.7 | 51.1 |
| elasticsearch | REST Search Engine | 31.7 | 31.5 | 31.8 | 31.9 | 32.1 |
| gradle | Build System | 36.3 | 39.8 | 40.1 | 39.6 | 39.5 |
| hadoop-common | Map-Reduce Framework | 38.4 | 41.7 | 42.5 | 40.9 | 41.4 |
| hibernate-orm | Object/Relational Mapping | 58.7 | 59.8 | 59.9 | 59.3 | 59.9 |
| intellij-community | IDE | 33.8 | 36.8 | 36.1 | 35.3 | 35.1 |
| liferay-portal | Portal Framework | 65.9 | 66.1 | 68.2 | 65.4 | 66.4 |
| presto | Distributed SQL Query Engine | 46.7 | 47.9 | 48.3 | 47.1 | 47.6 |
| spring-framework | Application Framework | 36.8 | 38.3 | 38.6 | 38.5 | 39.3 |
| wildfly | Application Server | 44.7 | 45.0 | 45.9 | 45.2 | 45.2 |

Table 2: Rank 1 F1 scores for each of the embeddings.

Figure 1 shows the validation losses achieved on all 10 Java projects. Randomly initialized embeddings are shown in purple, Java embeddings in green, C in blue, Python in red and English in orange. It can be seen that for most projects the pre-trained embeddings train faster, achieve lower losses, are more stable and over-fit less.

Table 3 shows overlap, speedup and improvement in test loss for each project-embedding combination compared to random embeddings. Overlap is the percentage of tokens in the project vocabulary that also appear in the embedding vocabulary. Speedup is calculated as:

$$S = N_r/N_e$$

$N_r$ is the number of epochs taken by the random embedding to reach its best validation loss and $N_e$ is the number of epochs taken by a non-random embedding to reach that same validation loss.

Improvement is calculated as:

$$I = L_r/L_e$$

$L_r$ is the test loss achieved using random embeddings and $L_e$ is the test loss achieved using a non-random embedding. An $I$ of 1.05 would indicate a 5% relative performance improvement in test loss over random embeddings.

| Project Name | C | | | Java | | | Python | | | English | | |
| | Overlap | Speedup | Improvement | Overlap | Speedup | Improvement | Overlap | Speedup | Improvement | Overlap | Speedup | Improvement |
|---|---|---|---|---|---|---|---|---|---|---|---|---|
| cassandra | 0.54 | 2.2 | 1.11 | 0.58 | 1.8 | 1.09 | 0.52 | 2.75 | 1.1 | 0.53 | 2.75 | 1.11 |
| elasticsearch | 0.38 | 0.43 | 0.99 | 0.45 | 0.75 | 1 | 0.36 | 1.1 | 1 | 0.38 | 1.0 | 1.0 |
| gradle | 0.63 | 4.5 | 1.13 | 0.74 | 4.5 | 1.14 | 0.64 | 4.5 | 1.14 | 0.65 | 4.49 | 1.14 |
| hadoop-common | 0.38 | 1.14 | 1.01 | 0.4 | 1.6 | 1.01 | 0.33 | 1.6 | 1.01 | 0.34 | 1.95 | 1.01 |
| hibernate-orm | 0.46 | 1.6 | 1.06 | 0.54 | 0.8 | 1.04 | 0.48 | 1.33 | 1.03 | 0.52 | 1.51 | 1.06 |
| intellij-community | 0.33 | 1.17 | 1.05 | 0.4 | 2.33 | 1.05 | 0.32 | 1.75 | 1.04 | 0.36 | 1.41 | 1.07 |
| liferay-portal | 0.37 | 0.75 | 1.01 | 0.46 | 1.2 | 1.04 | 0.37 | 1.0 | 1.02 | 0.39 | 1.50 | 1.03 |
| presto | 0.54 | 2.0 | 1.05 | 0.62 | 1.71 | 1.04 | 0.54 | 2.0 | 1.04 | 0.54 | 2.41 | 1.05 |
| spring-framework | 0.36 | 1.5 | 1.05 | 0.46 | 2.0 | 1.04 | 0.35 | 1.5 | 1.05 | 0.37 | 1.5 | 1.05 |
| wildfly | 0.5 | 2.8 | 1.05 | 0.56 | 2.8 | 1.05 | 0.48 | 2.0 | 1.06 | 0.51 | 2.71 | 1.06 |

Table 3: Results relative to random embeddings for each project-embedding combination. Overlap is the percentage of tokens within the embedding vocabulary that also appear in the project. Speedup is relative speedup of convergence compared to random embeddings. Improvement is relative improvement in test loss compared to random embeddings.

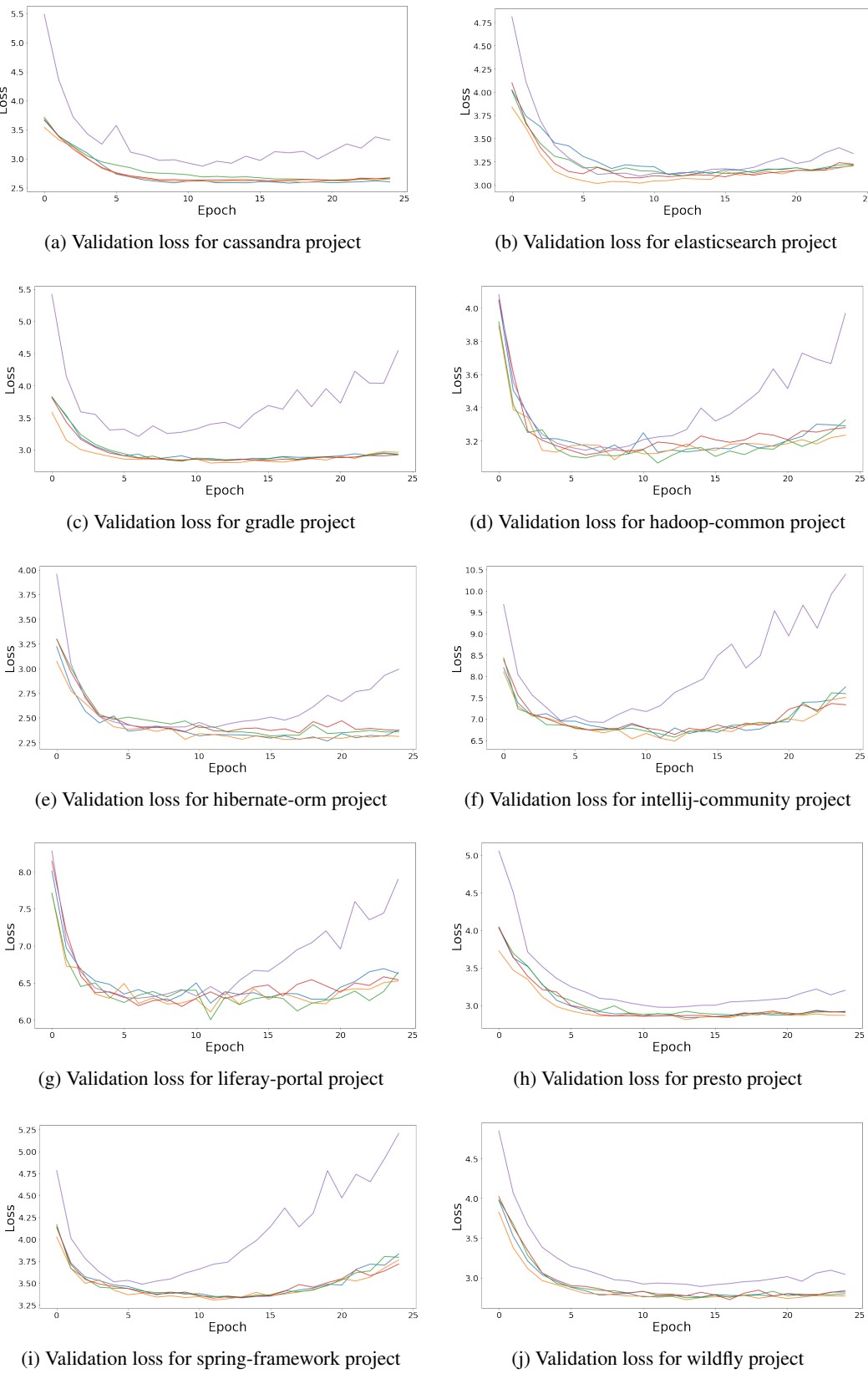

Figure 1: Validation losses for each of the 10 projects, averaged across 5 runs. Randomly initialized embeddings shown in purple, Java in green, C in blue, Python in red, and English in orange.

We notice that some projects, particularly the elasticsearch project (figure 1b), did not achieve any benefits from using the pre-trained embeddings and in fact experienced a slow down when using the C embeddings. To explore this further, we plotted speedup and improvement against overlap in figure 2.

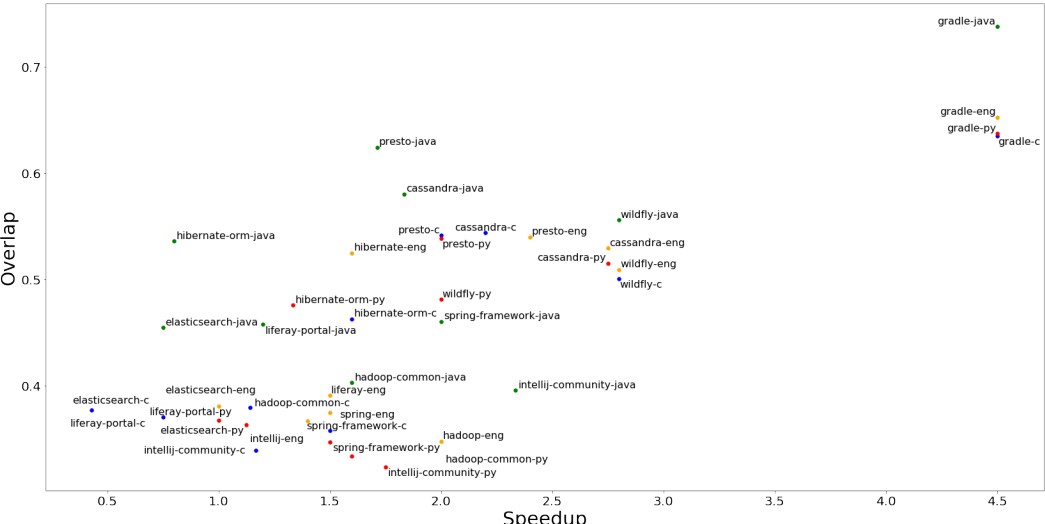

(a) Overlap vs. speedup over random embeddings. Correlation coefficient of 0.72.

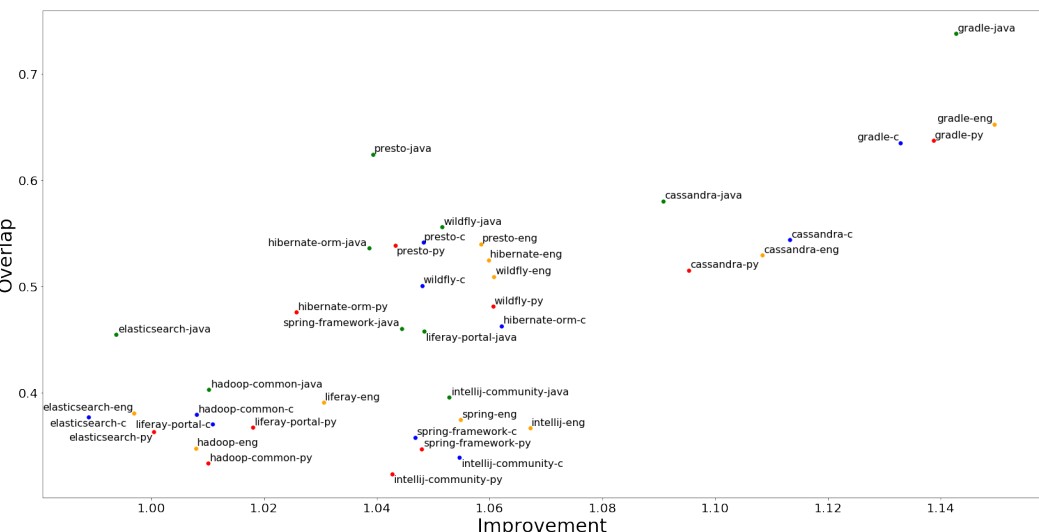

(b) Overlap vs. improvement over random embeddings. Correlation coefficient of 0.73.

Figure 2: Overlap vs. speedup/improvement on all project-embedding combinations. Points are colored: green for Java embeddings, blue for C embeddings, red for Python embeddings and orange for English embeddings.

We measure the Pearson correlation coefficient of each, receiving a coefficient of 0.73 for speedup and 0.73 for improvement, a medium to strong positive correlation for each. This implies that using more of the pre-trained embeddings provides more of both the speedup and improvement in test loss benefits.

Table 4 shows the overlap, speedup and improvement averaged over all projects. From this, we can see that we achieve around 1.9x speedup, along with a 5% relative improvement in test loss.

| Embedding | Overlap | Speedup | Improvement |
|---|---|---|---|
| C | 0.48 | 1.84 | 1.05 |
| Java | 0.54 | 1.98 | 1.05 |
| Python | 0.48 | 1.98 | 1.05 |
| English | 0.49 | 1.98 | 1.05 |

Table 4: Results relative to random embeddings for each of the pre-trained embeddings, averaged across all 10 projects.

Intuitively, it would make sense that the Java embeddings give the best results as the summarization task is also in Java. We see this is not the case, and the Java embeddings have the same speedup and improvement as the Python embeddings and only a small speedup improvement over the C embeddings, even though the average overlap using the Java embeddings is higher. Most interesting is the fact that the English embeddings achieve the comparable speedup and performance improvement, even though they have only been trained on human languages.

One potential reason for the similar performance between the embeddings that are trained on programming languages is that even though C, Java and Python are syntactically different, the extreme summarization task does not require much of this syntactic information. Consider the examples in table 5, which are Python versions of the Java examples in table 1. Although the syntax has changed (dynamic typing, no braces or semicolons, etc.) the available semantic information from the method body has not. This would imply that the language of the dataset used to pre-train code embeddings does not matter as much as the quality of the dataset with regards to sensible method and variable names. This is further solidified in the fact that the English embeddings, which have only been trained on human languages, achieved similar performance compared to programming languages.

```python
def get_surface_area (radius):
        return 4 * math.pi * radius * radius
```
```python
def get_aspect_ratio (height, width):
        return height / width
```

Table 5: Python versions of the Java examples from table 1

We also look at the amount of over-fitting on each project. From figures 1c, 1d, 1f, 1g and 1i we can see how the random embeddings show a large amount of over-fitting compared to the pre-trained code embeddings. We measure how much a project over-fits as:

$$O = L_b/L_f$$

$L_b$ is the best validation loss achieved and $L_f$ is the final validation loss achieved. We dub this term the over-fit factor, where an $O = 1$ would imply the final loss is equal to the lowest loss and thus has not over-fit at all (this could also mean the model is still converging, however from figure 1 we can see all project-embedding combinations converge before 25 epochs).

Table 6 shows the over-fit factors for each project-embedding combination. We can see that the random embeddings show the worst performance on every project. Interestingly, there appears to be no correlation between the overlap and the amount of over-fitting.

Table 7 shows the over-fit factor averaged across all projects for each embedding. Again, the results for each of the pre-trained embeddings are similar, showing that the language of the dataset used to train the embeddings does not have a significant impact on performance.

| Project | Random Over-fit | C Over-fit | Java Over-fit | Python Over-fit | English Over-fit |
|---|---|---|---|---|---|
| cassandra | 0.87 | 0.99 | 0.98 | 0.98 | 0.98 |
| elasticsearch | 0.91 | 0.97 | 0.96 | 0.95 | 0.96 |
| gradle | 0.74 | 0.97 | 0.97 | 0.97 | 0.97 |
| hadoop-common | 0.8 | 0.94 | 0.95 | 0.95 | 0.96 |
| hibernate-orm | 0.8 | 0.95 | 0.98 | 0.99 | 0.98 |
| intellij-community | 0.67 | 0.85 | 0.87 | 0.91 | 0.88 |
| liferay-portal | 0.8 | 0.94 | 0.9 | 0.94 | 0.92 |
| presto | 0.93 | 0.98 | 0.98 | 0.98 | 0.98 |
| spring-framework | 0.67 | 0.87 | 0.88 | 0.9 | 0.86 |
| wildfly | 0.95 | 0.97 | 0.98 | 0.97 | 0.98 |

Table 6: Over-fit factor for each project-embedding combination. Higher is better.

| Embedding | Over-fit |
|---|---|
| Random | 0.81 |
| C | 0.94 |
| Java | 0.95 |
| Python | 0.95 |
| English | 0.95 |

Table 7: Over-fit factor for each embedding averaged across all 10 projects. Higher is better.

## 5 RELATED WORK

Language models have been used as a form of transfer learning in natural language processing applications with great success (McCann et al., 2017; Howard & Ruder, 2018; Peters et al., 2018).

There has also been recent work on the further analysis of language models (Merity et al., 2018; 2017) and how well they assist in transfer learning (Mou et al., 2016).

The use of probabilistic models for source code originated from Hindle et al. (2012). From that, work on language models of code began on both the token level (Nguyen et al., 2013; Tu et al., 2014) and syntax level (Maddison & Tarlow, 2014).

Predicting variable and method names has become a common task for machine learning applications in recent years. Initial work was on the token level (Raychev et al., 2014; 2015; Allamanis et al., 2015) but it is beginning to become more common to represent programs as graphs using their abstract syntax tree (Allamanis et al., 2017b; Alon et al., 2018a;b).

## 6 DISCUSSION & CONCLUSIONS

We refer back to our research questions.

**Do pre-trained code embeddings reduce training time?** Yes, tables 3 and 4 show we get an average of 1.93x speedup. This is correlated with the amount of overlap between the task vocabulary and the embedding vocabulary, shown in figure 2a.

**Do pre-trained code embeddings improve performance?** Yes, tables 3 and 4 show we get an average of 5% relative validation loss improvement. Again, this is correlated with the amount of overlap between the vocabularies, shown in figure 2b.

**Do pre-trained code embeddings increase stability of training?** Although this is difficult to quantify due to how over-fitting interacts with the variance of the validation loss curves, from figures 1a and 1c we can see a clear increase in the variance of the validation loss curves using random embeddings compared to those using pre-trained embeddings.

**Do pre-trained code embeddings reduce over-fitting?** Yes, tables 6 and 7 show that the random embeddings over-fit more than the pre-trained embeddings on every project. However, this does not seem to have a correlation with the amount of vocabulary overlap and further work is needed to determine the cause of this.

**How does the choice of corpora used for the pre-trained code embeddings affect all of the above?** Intuitively, it would seem the best pre-trained embeddings would be those that are trained on the same language as that of the downstream task, but this is not the case. We hypothesize through the examples shown in tables 1 and 5 that the differing syntax between the languages is not as important as sensible semantic method and variable names within the dataset. This semantic information is also contained in human languages, which explains why the English embeddings also receive comparable performance.

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
