# OpenReview forum: "The Effectiveness of Pre-Trained Code Embeddings"
_ICLR.cc/2019/Conference_

### Official Review · AnonReviewer1 · 2018-11-02
**Are pre-trained code embeddings more effective than pre-trained natural language embeddings?**

**Rating:** 5
**Confidence:** 4

**Review:**

THE EFFECTIVENESS OF PRE-TRAINED CODE EMBEDDINGS

Summary:

This work shows how pre-training word vectors using corpuses of code leads to representations that are more suitable than randomly initialized (and trained) representations for function/method name prediction (here called extreme summarization). This paper applies a standard language model to several collected corpuses of code written in different programming languages in order to pretrain the embeddings. It then uses a standard model for the extreme summarization task that takes pre-trained language model embeddings. This leads to speedups in training, improvements in validation loss, and less overfitting. I worry that these results didn't really need much proving given that we have already seen the exact same methods work with natural languages. It would actually be more surprising if they didn't work for programming languages, which suggests that the real question is whether code embeddings are actually more effective than natural language embeddings for this problem given that the authors show syntax of the code is far less important than the semantics of the words in the vocabulary. It is not clear that embeddings pre-trained with much more data on natural languages wouldn't work just as well.

Pros:

This paper takes the time to clearly explain the effectiveness of pre-training words vectors in a new setting. It is easy to follow and understand thanks to the clear organization and exposition.

Speedup and validation loss improvements are demonstrated for a variety of programming languages despite the final evaluation being only in Java, which is quite surprising.

The authors discover that overlap in actual programming language syntax is less important than overlap in semantic representation.

Because the models are out-of-the-box, it is easy to focus on the actual contributions of this paper related to pre-training.

Does seem to clearly demonstrate that pre-trained embeddings of some form should be used for this extreme summarization task.


Cons:

It is not clear how big the collected corpus is. This is important because the work that this paper cites on pre-trained embeddings (Mikolov et al 2013, Pennington et al 2014, McCann et al 2017, Peters et al 2018) typically use fairly large datasets for pre-training. All of these results might be watered down by insufficient pre-training data for the language model when in fact the results could be much stronger with more data. It would be nice to show the effects of pre-training dataset size as is done in the aforementioned previous works. Without this comparison, it is hard to tell whether the paper sufficiently explores this idea.

The models are both standard, out-of-the-box models. There is no novelty on the modeling side of this paper.

The pre-training methods are also not novel. They are methods that have already been shown to work applied in a slightly different setting.

It is not clear that the setting is actually different enough to require this pre-training. Comparing to randomly initialized embeddings is fine, but I would also like to see a comparison to other pre-trained embeddings like GloVe, GloVe+CoVe, or ELMo (Pennington et al 2014, McCann et al 2017, Peters et al 2018). Since the authors find that it is the semantics of the words that matter more than the syntax of any particular programming language, then perhaps it would actually be better to use pre-trained embeddings that tap into much larger amounts of data. At the very least, it seems it would make sense to perhaps supplement a standard pre-trained embedding with those suggested by the authors since so many of the words in the code must be English words. If this is too farfetch'd, then I would suggest that the authors provide some statistics showing why GloVe, GloVe+CoVe, and ELMo are not appropriate starting points for comparison, but the overlap from the pre-training corpuses is already so low that it seems supplementing with standard pre-trained embeddings should only help.

The evaluation dataset detailed in Allamanis et al 2016 uses two metrics: an F1 metric and an exact match metric. This paper only compares on validation loss. What's more, it reports everything in relative terms so that the raw improvement is masked until Figure 1 makes it somewhat possible to deduce. The problem here is that we don't know how a 0.0-0.5 raw improvement in validation loss translates to the metrics established for the dataset by Allamanis et al 2016. If those are no longer the standard metrics, the authors should explain how validation loss came to supplant the original metrics proposed by Allamanis et al 2016.

What's more, there is no context for how well models typically do on this evaluation task. Without any comparisons it is impossible to tell whether any of the experiments are using models in a reasonable realm of performance on this task.

Overall:

All these effects have already been shown for pre-trained embeddings in the past, and the experiments involve running standard methods on newly collected datasets. This means there is no novelty in the pre-training method or the extreme summarization method. Little is known about the newly collected datasets, it is not clear how to interpret the relative improvements in validation loss compared to the original metrics of Allamanis et al 2016, and the paper lacks necessary comparisons to othef pre-trained embeddings, so though the overall claim that pre-trained embeddings should be used for this task seems to hold up, it is not clear that this is a complete argument for the method chosen by the authors.

---

> ### Author Response · Authors · 2018-11-12
> **Reviewer 1 response**
>
> Thank you for the comments and feedback.
>
> 1. We have added the number of tokens and lines of code within the embedding datasets to section 3.1. We agree with your assumption that more data for pre-training would most likely be beneficial, however we believe this is outside the scope of this work as we have aimed to present an initial exploration into the effectiveness of pre-trained code embeddings. Now we have shown that embeddings do provide several benefits, further work can be done on exploring how best to create these embeddings, e.g. more data, which algorithms to use, etc.
>
> 2. Thank you for your comment suggesting the use of embeddings pre-trained on natural languages. It encouraged us to explore this, and we have obtained results using embeddings created from the WikiText-103 dataset, using the same AWD-LSTM-LM model. We used this dataset as it is of comparable size to the programming language data. Additionally, we believed that using GloVe (or other) pre-trained embeddings would raise questions of whether the larger amount of data the GloVe embeddings have been trained on is responsible for the results. We have included this within our paper, which now indicates that the natural language (English) embeddings show comparable results to the programming language embeddings, thus reinforcing our view that the semantics are more important than the syntax for this task (see also our response to AnonReviewer2).
>
> 3. We initially stated that the "improvement" metric was in terms of validation loss. This was a mistake and should have said test loss. The paper has been updated to fix this and "test loss improvement" has now been explicitly stated throughout.
>
> 4. We have now updated our paper to include mentions of rank 1 F1 scores throughout the paper and have added a table of F1 scores. We can now see that a 4\% test loss improvement corresponds to a 5\% rank 1 F1 improvement.

---

> > ### Comment · AnonReviewer1 · 2018-11-26
> > **Thank you for your additional work**
> >
> > Thank you especially for doing such a nice job of supplying additional experiments that show a comparison to pre-trained English embeddings. The results appear to be in keeping with your observations about syntax vs. semantics as well as my expectations. I'm slightly disappointed that they did show that pre-trained code embeddings are not any more effective than pre-trained English embeddings, but it is interesting that with comparable data, some programming languages are providing pre-trained embeddings as useful as those trained on English.

---

### Official Review · AnonReviewer2 · 2018-11-03
**Interesting and important research questions, unconvincing results**

**Rating:** 4
**Confidence:** 4

**Review:**

This paper sets to understand whether pretraining word embeddings for
programming language code by using NLP-like language models has an
impact on extreme code summarization task (i.e., generate/predict the
name of a function based on its body).

I think the paper asks some important questions, however the execution
of the research and the results presented are not convincing.

I think the area is relevant and the research questions are worth
pursuing; however the work as it is presented in the paper needs
improvement to be accepted for publication.

Pros:
* The study of language models for programming language code
* Pretraining is performed for 3 different languages (C, Java, Python) - target task is in Python

Cons:
* Strange claims of speedup and performance improvement
* Inconclusive results

Some suggestions for improvement:

* The section on language models pretraining is very sparse, more
  details are needed.

* The claims of speedup and improvement are strange. Speedup refers to
  the training speed, I suppose. The performance of the downstream
  task is never discussed. Only the validation loss is shown and all
  the performance "improvement" is discussed on these graphs, which I
  found strange. Also, the graphs have their y-axes starting at
  non-zero values. I personally prefer graphs that start at zero and
  if there is a need to "zoom-in" find a way to "zoom-in" to the part
  of the graph that is important.

* In general the paper writing and reporting on the experiments sounds
  ad-hoc and not well thought-out.

* I don't agree with many of the explanations in the paper. For
  example (page 6), it's not true that the extreme summarization task
  does not require much of the syntactic information (there are
  submission at the current ICLR'19 that show exactly the opposite,
  encoding based on syntactic information is useful). The model
  studied in the paper does NOT use any syntactic information, it
  treats the code like a sequence of tokens.

* The last question in Section 6 is not a Yes/No question, the answer
  is phrased as a Yes/No question.

I encourage the authors to pursue the research questions, however in a
more systematic and with better methodology.

---

> ### Author Response · Authors · 2018-11-12
> **Reviewer 2 response**
>
> Thank you for the comments and feedback.
>
> 1. We have added details about the size of the datasets for the language modeling. More details about the model are available in the papers by Merity et al., which is referenced in our paper.
>
> 2. You are correct in that speedup refers to the increase in training speed. This is detailed in the results section where we mention how speedup, S, is calculated as the number of epochs taken by the random embeddings to reach its best validation loss, N\_r, divided by the number of epochs taken by a non-random embedding to reach that same validation loss, N\_e.
>
> 3. The downstream task in this experiment is the extreme summarization task. We have clarified this in the abstract \& introduction.
>
> 4. The performance improvement is for the test loss (using the parameters achieved from the lowest validation loss). The paper previously incorrectly stated that improvement was calculated via the validation loss and this has now been corrected. We have also added a table of results for the rank 1 F1 scores and made sure to explicitly mention the performance improvements (both for F1 scores and test loss) in the results section.
>
> 5. We agree that syntactic information is useful and you are correct in that the model is not explicitly fed the syntactic information. However, the model does implicitly use syntactic information as the syntax does exist within the sequence of tokens. Furthermore, as per AnonReviewer1's comments, we have added results obtained from using embeddings trained on a natural language (English), and have shown they achieve comparable results to each of the programming languages used. As all 4 of these embeddings achieve similar results - with the main difference between them being the syntax - we would argue that this supports the view that the syntax is less important - although admittedly still useful - than the semantic information provided by sensible variable names for this task. We do think the usefulness of syntactic information for the extreme summarization task is an interesting area of research and requires further investigation, but we believe it is outside the scope of this work.
>
> 6. Thank you for spotting the error with RQ6, this has now been corrected.

---

### Official Review · AnonReviewer3 · 2018-11-05
**Incremental empirical study**

**Rating:** 6
**Confidence:** 3

**Review:**

The paper presents some experiments using pre-trained code embeddings on the task of predicting a method name from the code of method body. The paper is well written and the motivations and the design of empirical study are clear.

The empirical results of validation loss in Figure 1 is reporting the behaviour of random initialization of embedding. From the plots of 10 projects we may derive a couple of claim: (i) the validation loss of random initialization after 10 epochs may increase and get unstable, (ii) random initialization after 5-10 epochs may reach the same loss as pre-trained embeddings. The working assumption is that pre-trained embedding should speed-up the learning process. The empirical results show that it is not just a matter of reducing the training time but also of performance. The discussion is neglecting to comment this behaviour that looks not compliant with the working assumptions.

Minor comment. The reference [Allamanis et al., 2016] is pointing to arxiv.org despite the fact that the work is published as Proceedings of the 33 rd International Conference on Machine Learning, New York, NY, USA, 2016. JMLR: W&CP volume 48.

---

> ### Author Response · Authors · 2018-11-12
> **Reviewer 3 response**
>
> Thank you for the comments and feedback.
>
> 1. To clarify, the working assumptions is that pre-trained embeddings do increase performance. This is mentioned in the abstract - ''which provide benefits such as reduced training time and overall performance" - and in the fourth paragraph of the introduction, with a reference to (Kim, 2014).
>
> 2. We have now fixed the reference, thank you for spotting that.

---

### Author Response · Authors · 2018-11-12
**Revisions**

We would like to thank all reviewers for their thoughtful comments.

We have made the following revisions to the paper:

1. Clarified that extreme summarization is the downstream task
2. Added results obtained from comparing natural language (English) embeddings to the programming language embeddings
3. Added information about the size of the embedding data
4. Added rank 1 F1 score results
5. Fixed where the improvement metric was incorrectly stated to have been calculated from the validation loss
6. Fixed a handful of typographical errors
7. Fixed a reference

---

### Meta-Review · Area_Chair1 · 2018-12-13
**Reasonable experiments, but limited contributions**

**Confidence:** 3
**Recommendation:** Reject

**Metareview:**

All three reviewers agree that the research question—should pretrained embeddings be used in code understanding tasks—is a reasonable one. However, there were some early issues with the way in which the paper reported results (involving both metrics and baselines). After some discussion with the reviewers, it seems that the paper now presents a clear picture of the results, but that these results are not sufficiently strong to warrant acceptance.

I'm wary to turn down a paper over what are basically negative results, but for results like this to be useful to the community, they'd have to come from a very thorough experiment, and they'd have to be accompanied by a frank and detailed discussion. Neither of the two more confident authors are convinced that this paper meets that bar.